# Evaluation of a novel real-time adaptive assist-as-needed controller for robot-assisted upper extremity rehabilitation following stroke

Ana P. Arantes[1], Nadja Bressan[2], Ludymila R. Borges[3], Chris A. McGibbon[4]*

1 Hotchkiss Brain Institute (HBI), Cumming School of Medicine, University of Calgary, Calgary, Canada,
2 Faculty of Sustainable Engineering Design, University of Prince Edward Island, Charlottetown, Canada,
3 Assistive Technology Laboratory (NTA), Faculty of Electrical Engineering, Federal University of Uberlândia, Uberlândia, Brazil, 4 Institute of Biomedical Engineering and Faculty of Kinesiology, University of New Brunswick, Fredericton, Canada

* cmcgibb@unb.ca

**Data Availability Statement:** Data are now available at https://dataverse.lib.unb.ca/dataset.xhtml?persistentId=doi:10.25545/OBF93E

## Abstract

Rehabilitation therapy plays an essential role in assisting people with stroke regain arm function. Upper extremity robot therapy offers a number of advantages over manual therapies, but can suffer from slacking behavior, where the user lets the robot guide their movements even when they are capable of doing so by themselves, representing a major barrier to reaching the full potential of robot-assist rehabilitation. This is a pilot clinical study that investigates the use of an electromyography-based adaptive assist-as-needed controller to avoid slacking behavior during robotic rehabilitation for people with stroke. The study involved a convenience sample of five individuals with chronic stroke who underwent a robot therapy program utilizing horizontal arm tasks. The Fugl-Meyer assessment (FM) was used to document motor impairment status at baseline. Velocity, time, and position were quantified as performance parameters during the training. Arm and shoulder surface electromyography (EMG) and electroencephalography (EEG) were used to assess the controller's performance. The cross-sectional results showed strong second-order relationships between FM score and outcome measures, where performance metrics (path length and accuracy) were sensitive to change in participants with lower functional status. In comparison, speed, EMG and EEG metrics were more sensitive to change in participants with higher functional status. EEG signal amplitude increased when the robot suggested that the robot was inducing a challenge during the training tasks. This study highlights the importance of multi-sensor integration to monitor and improve upper-extremity robotic therapy.

## Introduction

Stroke is a leading cause of adult long-term severe disability and the second cause of mortality worldwide [1]. Eight out of ten people with stroke have chronic hemiparesis, which results in impaired motor control of the affected (contralesional) side [2]. Chronic hemiparesis interferes with daily function, where once simple reaching and repositioning tasks become difficult

**Funding:** APA: Coordenacao de Aperfeicoamento de Pessoal de Nivel Superior, Brasil (CAPES), #001 - The funders had no role in study design, data collection and analysis, decision to publish, or preparation of the manuscript. CM: New Brunswick Innovation Foundation, Research Assistantship Initiative, #2019-046 -The funders had no role in study design, data collection and analysis, decision to publish, or preparation of the manuscript.

**Competing interests:** The authors have declared that no competing interests exist.

and frustrating. Many hours of manual therapy are required to treat chronic hemiparesis, and results can vary according to the degree of residual function post-stroke [3].

Upper-extremity robotic rehabilitation of chronic hemiparesis has shown some promise in improving motor recovery post-stroke. Although not readily available in clinics, the use of this technology in the rehabilitation field is widely thought to be potentially beneficial [4–14]. Several studies have shown how the strategy of control used in robot-assisted rehabilitation can significantly and positively impact the rehabilitation process's effectiveness and efficiency [4, 5, 7, 9–14], but also highlight drawbacks such as "slacking" where the user lets the robot drive their movements without performing sufficient physical effort to benefit from the training. Indeed Zhang et al. [15] recently reported on an assist-as-needed (AAN) upper extremity rehabilitation robot addressing this fact, where the device aids when the participants were unable to accomplish the task. However, a comprehensive examination of the participants' advancement of both muscular and cerebral activity, which is a crucial component to assess motor recovery, was absent.

Importantly, prior studies also do not incorporate adaptation of the controller based on real-time physiological performance that can appropriately scale assistance or resistance to improve engagement and progression [5, 9, 11–14]. As presented in Table 1, to our knowledge, no studies have validated this approach or integrated the capability of challenging the users to keep them engaged. Past research shows this is a requirement to promote motor plasticity and faster recovery [6–8, 11, 12, 15].

The present study aimed to develop and validate a novel adaptive AAN (aAAN) algorithm for upper-extremity robot-assisted therapy for stroke. The controller facilitates the completion of tasks for users while providing just enough force to support their efforts and maintain motivation, while also mitigating the potential for complacency. A key feature of our system is its adaptability, wherein it responds to the user's individual requirements by gradually reducing assistance and increasing task difficulty as the user's performance improves. This tailored approach aims to optimize the training experience and foster engagement through non-monotonic adjustments.

This new algorithm's control system integrates EMG and a haptic system to provide a subject-cooperative, non-slacking, robotic-assisted therapy to facilitate and accelerate recovery of upper extremity motor function. Changes in performance metrics, EMG entropy amplitude, and EEG power density between the baseline and training sessions were examined for each participant. Relationships of these responses to participants' functional status using the Fugl-Meyer (FM) clinical assessment at baseline were explored to describe the robot training system's sensitivity, allowing us to quantify how performance metrics would be expected to change with improvement in functional status in future interventional studies.

**Table 1. Previous assisted- as- needed upper extremity robotic-rehabilitation system for stroke patients.**

| Authors | EMG Based Algorithms | EEG Based Algorithms | Motion/ Force Based Algorithms | EEG As Assessment Tool |
|---|---|---|---|---|
| LEONARDIS ET AL. [16] (2015) | x | | | |
| DIPIETRO ET AL. [17] (2005) | x | | x | |
| BALASUBRAMANIAN ET AL. [18] (2018) | x | x | | |
| HU ET AL. [19] (2013) | x | | | |
| LOSEY ET AL. [20] (2020) | | | x | |
| BHAGAT ET AL. [21] (2020) | | x | | x |
| CHEN ET AL. [22] (2022) | x | | x | |
| ZHANG ET AL. [15] (2020) | | | x | |
| MOUNIS ET AL. [23] (2019) | | | x | |

## Materials and methods

The following essential requirements were prescribed to develop robot-assisted training with an adaptive assist-as-needed system.

First, the robot shall provide enough force to allow the user to complete the task and motivate them but do so minimally to avoid the slacking effect. Second, the system shall adapt to the user's needs by, for example, reducing the assistance and increasing the challenge as the user's performance improves. Third, the training should be non-monotonic in order to be more engaging. These elements are expected to encourage voluntary control during training, known to promote motor plasticity in the brain [24, 25]. Finally, multiple sensors should measure movement performance and muscle and brain activity during the training to capture outcomes relevant to motor recovery.

The system was implemented using a PHANTOM 3.0 device from SensAble technologies (3D systems, Rock Hill, South Carolina), with open-source software that allows the user to interact with it in a broad range of applications. The PHANTOM device is a high-precision, low friction, instrument with a large workspace and capable of generating high forces with high fidelity. The device is mostly used for rehabilitation, the development of games, and entertainment. It can provide low-level control, support the visualization of 3D objects, and can generate force effects. Information such as velocity, force, and position can be provided from this device.

EMG was measured with the Delsys Trigno™ wireless system (Delsys, Inc., Natick, Massachusetts). Electrodes were placed on biceps (BI), triceps (TI), pectoralis major (PM), anterior deltoid (AD), middle deltoid (MD), and posterior deltoid (PD) of the affected arm, followed the Surface electromyography for the non-invasive assessment of muscle (SENIAM) guidelines.

EEG was measured with the Artinis system (Artinis Medical Systems, Einsteinweg, The Netherlands) which has 128 channels for full head coverage. The channels CF3, CF4, C3, C4, CP3, CP4, P3, P4, and Pz were used for this protocol following the 10/20 system configuration.

## Control algorithm

The information acquired in real-time from the PHANTOM and EMG systems was used to control the robot-assisted system. For the novel subject-cooperative non-slacking system, the system's initial trigger was regulated based on the EMG signal. The EMG control part of the system was similar to the approach described in Zhang et al. [15], where the sample entropy was computed for the EMG signal in order to detect the muscle activity onset. This approach can distinguish voluntary surface EMG signals from spurious background spikes which are common in resting paretic muscle.

The EMG signal of three muscles—deltoid, biceps, and triceps—was constantly analyzed. The velocity was also acquired continuously as soon as the EMG trigger threshold was first exceeded. Importantly, the robot only assisted the user when sufficient EMG has triggered the system and is maintained activated, but insufficient motion is detected to achieve the target. Fig 1 illustrates how this method works.

The system adapts to the participant's progress based on their EMG signal, the velocity, and force data they can generate throughout the training sessions. The force and velocity data were acquired from the Phantom device. In subsequent training sessions, the EMG threshold value progressively increases, and once the driving muscle can generate stronger EMG signals some resistance force is then gradually added to the system to provide a challenge for evoking cortical engagement. The initial threshold (buffer) was calculated based on the EMG collected during the baseline (robot-off) phase. The threshold was calculated as 20% of the maximum EMG

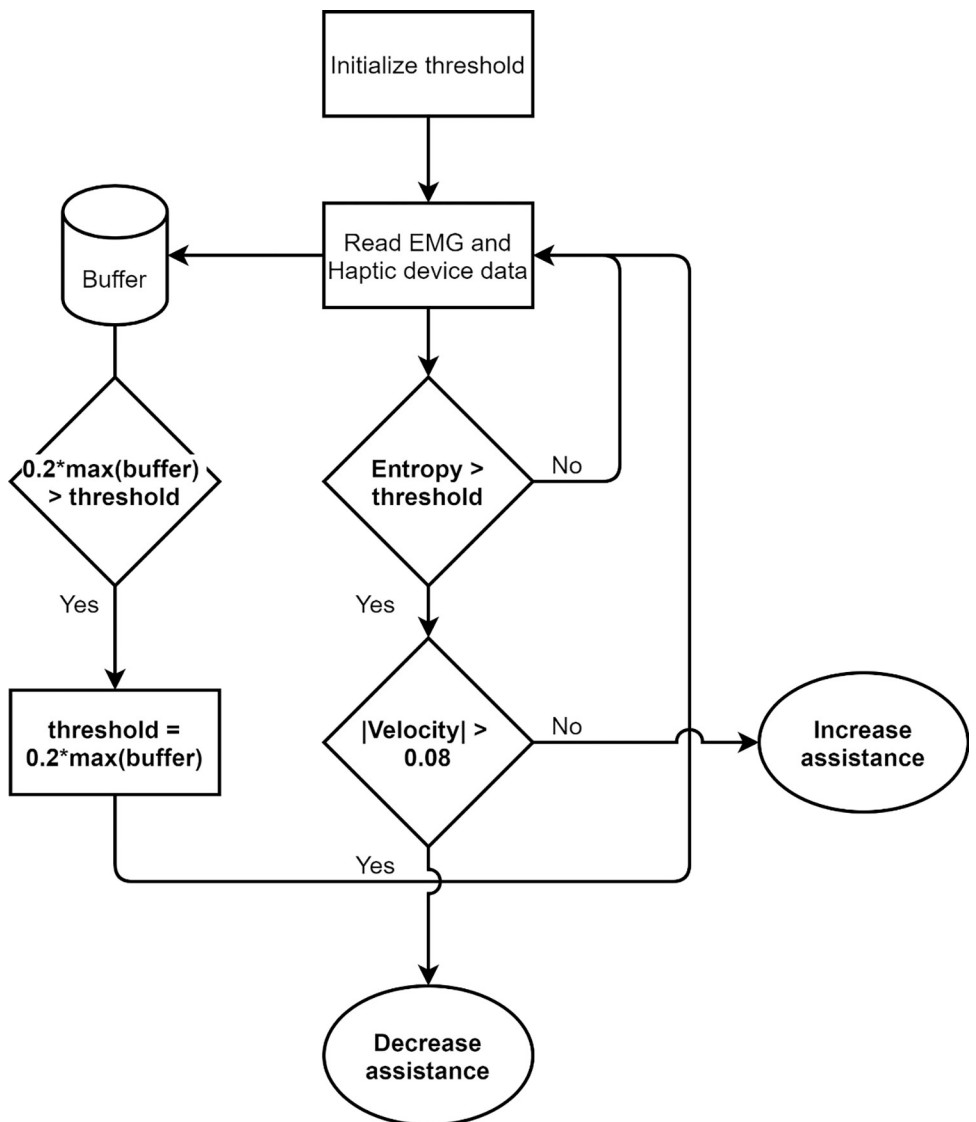

**Fig 1. Subject-cooperative non-slacking system.**

entropy amplitude. This threshold was adapted for each repetition during the training (robot-on) phase. This strategy was used to encourage the user's effort during training–helping at first when needed but challenging them as performance improves.

If this approach is effective, one would expect that those with higher disability at baseline (lower FM scores) would respond more in movement performance between targets, and those with lower disability at baseline (higher FM) would respond more in measures of synergistic muscle and cortical activity.

## Experiment

The study setting was a university research laboratory. The protocol was approved by Research Ethics Boards of the Regional Health Authority and the University where the research was conducted. The original planned design was a prospective cohort study of 10 stroke participants over 20 training sessions. Study recruitment began in February 2020 and ended in mid-

March 2020 due to the COVID pandemic shutdown. As such this study focuses on the baseline training sessions of the five participants recruited and tested prior to the national lockdown. All participants provided written informed consent after a full explanation of the research and the experiment.

**Participant's selection criteria.** Participants were community-dwelling adults recruited through the local rehabilitation hospital and stroke rehabilitation unit of the local primary care hospital. Individuals were first approached by their therapist (not involved in the study) and provided with contact information if they were interested in participating. To reduce bias, inclusion criteria were broad as possible: >19 years of age; chronic hemiplegic stroke affecting upper extremity, >6 months; able to actively move their shoulder, elbow, and wrist; Modified Ashworth Scale (MAS) [26] 3 or less for all joints.

Excluded were those who had: a history of severe neurological injuries other than or stroke; severe concurrent medical diseases; infections, circulatory, heart or lung, pressure sores; problematic spasticity (e.g., Modified Ashworth >3); severely compromised arm function (Fugl-Myer score < 21 on the upper-extremity scale); heterotopic ossification; significant contractures; psychiatric or cognitive situations that may interfere with the proper operation of the device; cognitive impairments resulting in an inability to follow directions; poor skin integrity in areas in contact with the device; uncontrolled Autonomic Dysreflexia; and functional visual field or hemispherical neglect. These exclusion criteria were selected based on previous research studies of robot-assisted therapy in a clinical population with impaired arms [7, 21, 23].

**Testing protocol.** The participant was seated in a height-adjustable chair with their arms in an initial position with the elbow at 90 degrees flexion on a tabletop with a monitor under the tabletop facing up. The PHANTOM device was positioned in front of them. Fig 2 shows a diagram of the setup and data collection.

The table was a transparent plexiglass acrylic sheet on top of a monitor mounted to a metal structure. The monitor showed the letters, one letter at a time. The participant was asked to hold the robotic arm's end effector and move until it touches the letters. They were able to feel when they reached the letter through a haptic force (touch effect) programmed in the robot.

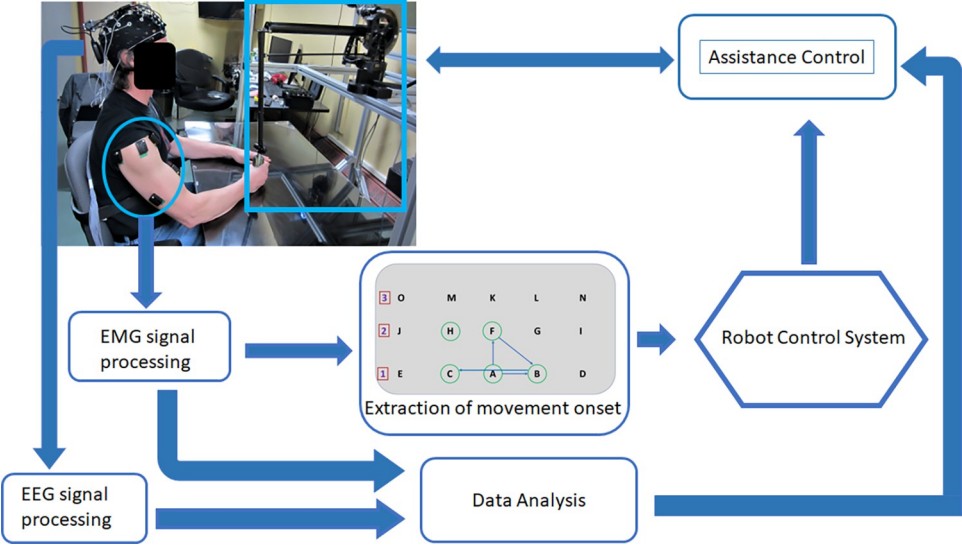

**Fig 2. Diagram of the algorithm used during the training phase.**

Prior to data collection, FM scores were acquired. The next step was to set up the EEG and EMG electrodes. Nine EEG electrodes and six EMG electrodes were placed on the participant according to Fig 2.

Six tests involving a planar 2 degree of freedom (DOF) reach task were performed: one for a baseline and five for the training. The data collection session lasted between 45 minutes to one hour. In the first protocol, called baseline, which was used to acquire baseline characteristics, the participant was asked to reach the letters appearing on the monitor. The letters appeared in alphabetical order, one at a time, three times. First, only the initial line of letters, A to E, show up on the screen. Simple horizontal tasks were evaluated with the elbow bent at this point. Next, the second line of letters, F to J show up, again to evaluate horizontal tasks but increasing the elbow extension. Finally, the last line of letters again show up to increase elbow extension and once again evaluate horizontal tasks. The letter positions can be seen in Fig 2.

After that, letters appeared in a sequence that the participant had to reach. For example, letter A shows first, then letter F, so the participant performs a reaching forward task with the elbow half extended. Next, letter B appears, then letter H requiring lateral reaching with elbow half extended, and so on. The participant is given 5 seconds to reach the target, if they are unable to get to the target within this time, the target disappears, and the next target shows up on the screen.

The baseline phase was used to collect range of motion (ROM), velocity, force, and electromyography signal (EMG) information. No assistance was provided from the robot during the baseline phase. Then, during subsequent training protocols, the participant had to repeat the same movements as the baseline phase for the following training protocols. However, this time, the letters appeared on the screen randomly. The letters appear 30 times for training tests one, two, and three, and 60 times for training tests four and five, with an interval 5 seconds between them. The participants had 60 seconds to rest between the tests.

In this part of the training the controller (as described above) dictates the protocol: The robot-arm assisted the participants when needed; if the participant could not reach the target, the robot-assisted the participant in getting to the target. The robot continuously evaluated the participant's EMG signal throughout the trajectory to achieve the target and only assisted if the EMG signal was still detectable and there was no movement. The robot also applied some resistance to challenge the participant if both the velocity and the EMG were higher than their respective thresholds. The EMG signal threshold necessary for the robot to assist/resist increased based on the participant's improvement.

## Data analysis

**Position metrics.**   Position parameters were calculated for each move between displayed letters, similar to the process proposed by Krebs et al. [27]. The first position parameter, $P_1$, represented the average distance between the PHANTOM handle and the target along the trajectory. If one takes longer to start moving and has difficulty moving to the target, $P_1$ increases. On the other hand, a quick and linear movement from the handle position to the target yields a lower value of $P_1$. Fig 3 shows an example of the $d_i$ measurements along a movement trajectory from letter A to letter B.

The second position measure, $P_2$, is the root-mean-square deviation of the normal distance between the handle and line between targets. In this equation, the reference is the smallest distance between the two targets. Hence, for someone with high dexterity and coordination, $P_2$ is small; otherwise, as shown in Fig 3, the value of $P_2$ increases.

The position parameters $P_1$ and $P_2$ were calculated using Eqs (1) and (2), where $dn_i$ is the normal distance to the line between targets as defined above, and the coordinates represent the

**Fig 3. Position parameters where $(x, y)$ is the target position and $(x_i, y_i)$ is the handle position, and N is the number of samples.**

position of the letter of origin $(x_a, y_a)$, the position of the target letter $(x_b, y_b)$, and the handle position $(x_i, y_i)$ at time frame index $i$.

$$P_1 = \frac{1}{N} \sum_{i=1}^{N} d_i \ where: \ d_i = \sqrt{(x - x_i)^2 + (y - y_i)^2} \tag{1}$$

$$P_2 = \sqrt{\frac{1}{N} \sum_{i=1}^{N} dn_i^2} \ where: \ dn_i = \frac{|(y_b - y_a)x_i - (x_b - x_a)y_i + x_b y_a - y_b x_a|}{\sqrt{(y_b - y_a)^2 + (x_b - x_a)^2}} \tag{2}$$

$P_1$ and $P_2$ values were then normalized by dividing this value by the assistance-resistance force (ranging from -6 to +6) and called $P_{1norm}$ and $P_{2norm}$. A negative force represents robot resistance, while a positive value represents assistance. Thus, higher values of $P_{1norm}$ and $P_{2norm}$ represent better performance.

**EMG mean entropy.** EMG entropy was calculated as described by others [28]. Because stroke participants are likely to have muscle spasticity which could add entropy to the signal that is unrelated to neural drive, EMG entropy was normalized to an index of spasticity. The spasticity index was calculated by summing ordinalized MAS [26] scores ("0" = 0, "1" = 1, "1+" = 2, "2" = 3, etc.) across wrist, elbow and shoulder, and rescaling between 1 and 10. This value was then used to calculate the adjusted EMG mean entropy.

**EEG activity.** EEG data was collected from channels CF3, CF4, C3, C4, CP3, CP4, P3, P4, and PZ. The data was first band-pass filtered from 1 to 30 Hz and visible artefacts were removed using an independent component analysis procedure to calculate the EEG spectra. Following this, the data were processed using a common average reference, and the data were divided into rest and active. The active and rest data were then divided into epochs of 2 seconds using a Hamming-window, and the power spectral density was calculated using the fast Fourier transform. The power spectral density was then averaged for each condition. To analyze brain activity differences between baseline (robot off) and training (robot on) sessions the beta wave range (16 - 24Hz) was selected which corresponds to attention and conscious focus.

**Statistical analysis.** EMG and EEG measurements were then compared between the robot off (baseline) and on (training) conditions using a paired samples t-test. Finally, curve fitting was used to quantify the relationship between baseline functional status (via the FM score) and training outcomes measures: position and velocity metrics, EMG, and EEG. Polynomials and rationales were used to find the simplest equation that best fit the experimental data. Curve fit data were then examined to describe the system's sensitivity to modify these outcomes in future prospective interventions.

**Table 2. Participant demographics, clinical scores, and unadjusted performance scores.**

| Subj. | Sex | Age | Side | FM | Spast. Score* | Mean EMG Robot Off | Mean EMG Robot On | Force Assist | $P_1$ | $P_2$ |
|---|---|---|---|---|---|---|---|---|---|---|
| 1 | F | 71 | L | 40 | 9.5 | 0.128 | 0.135 | 0.323 | -0.079 | 0.135 |
| 2 | M | 61 | R | 65 | 1 | 0.119 | 0.127 | -0.566 | -0.015 | 2.729 |
| 3 | M | 55 | R | 47 | 2 | 0.110 | 0.132 | 0.562 | -0.814 | 0.633 |
| 4 | M | 76 | L | 58 | 3 | 0.091 | 0.126 | 0.744 | 0.317 | 0.779 |
| 5 | M | 66 | R | 36 | 7 | 0.066 | 0.105 | 0.213 | 1.476 | 0.460 |

* Spasticity score = sum of MAS scores across wrist, elbow, and shoulder on 1–10 scale.

## Results

Five chronic post-stroke participants were enrolled in this pilot clinical study (Table 2).

To demonstrate the movement task and its relative difficulty for the participants, Fig 4 shows the hand horizontal trajectory for two participants: the highest functioning (FM = 65) and lowest functioning (FM = 36), during a task that required reaching a letter target and returning to the start letter.

As shown in Fig 4, the higher functioning participant moved smoothly between targets with only moderate difficulty returning to the start point. In contrast, the lower functioning participant had difficulty reaching the target and often did not return to the origin before moving to the next target.

### Effect of the aAAN algorithm

EMG Mean Entropy for each of the six muscles for the high and low functioning participants can be seen in Fig 5. The mean increase in normalized EMG entropy between baseline and training was statistically significant (t = 3.73, df = 4, p = .02), indicating that slacking was avoided regardless of functional status.

Although there was no statistically significant change in EEG spectral density between baseline and training (t = 1.18, df = 4, p = 0.31), Fig 6. shows that the system required more effort from less impaired participants but also suggests that the more impaired participants would require more training to demonstrate changes in cortical activity.

### Relationships to functional status

The relationship between functional status (FM score) and the various measures of interest (position metrics, velocity, EMG entropy and EEG) are presented in Fig 7. Best fit model and associated coefficients are summarized in the upper right corner of Fig 7. To summarize, position metrics $P_{1norm}$ and $P_{2norm}$ were fit with rational functions ($R^2 = 0.9955$ and $R^2 = 0.9996$, respectively) and $2^{nd}$ order polynomial models were used to predict average velocity ($R^2 = 0.9964$), EMG Mean Entropy ($R^2 = 0.8073$), and mean EEG ($R^2 = 0.9656$). The sensitive regions of each curve were then highlighted (blue shaded regions) for discussion.

## Discussion

Despite the great promise of robotic rehabilitation for improving motor function post-stroke, clinical adoption of the technology has lagged [29]. Although there are numerous contributing factors, one area of focus in recent years is how to avoid potentially confounding factors inherent in robot control, such as slacking behavior and disengagement of the rehabilitation patient [5, 9, 11–14]. To address this shortcoming, a novel EMG-based adaptive AAN algorithm was

### Training Session 4: FM=65

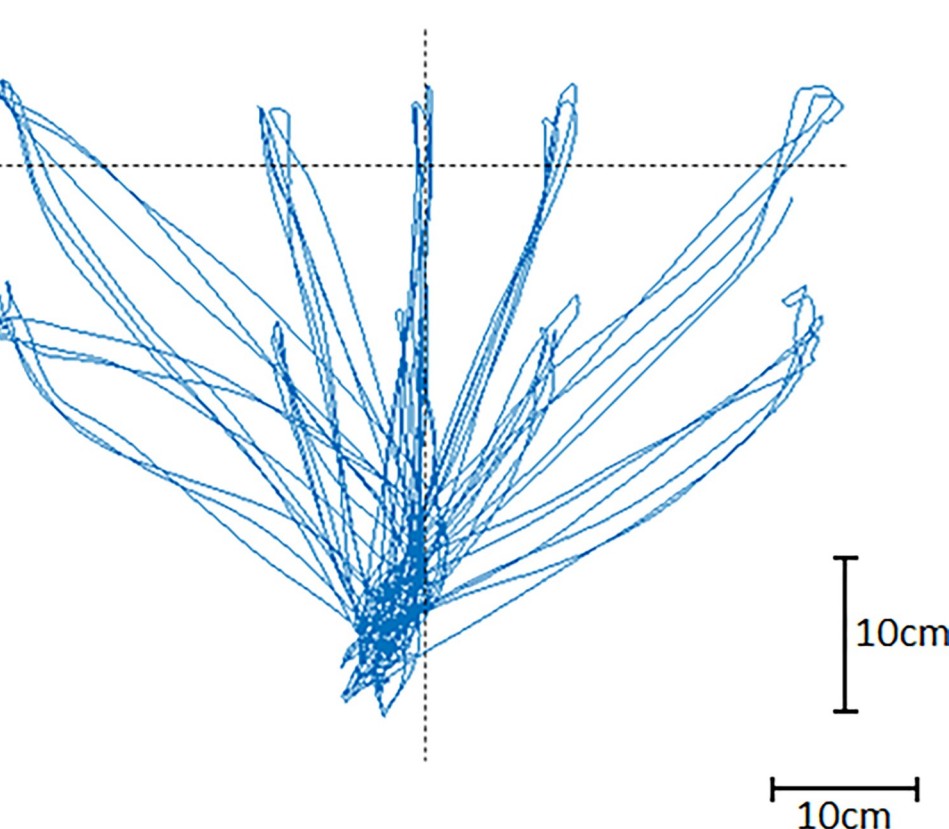

10cm

10cm

### Training Session 4: FM=36

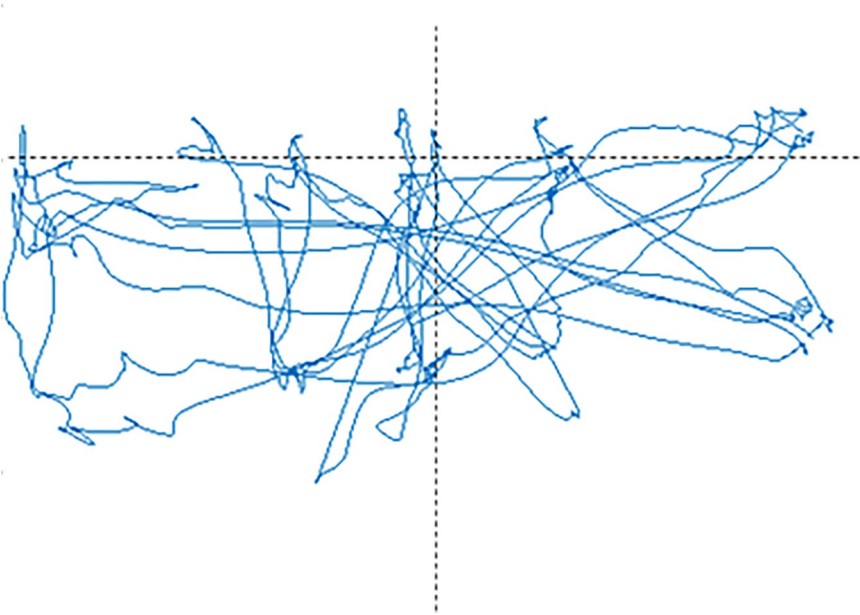

**Fig 4.** (a) Horizontal trajectory from the participant with FM score of 65; (b) Horizontal trajectory from the participant with FM score of 36.

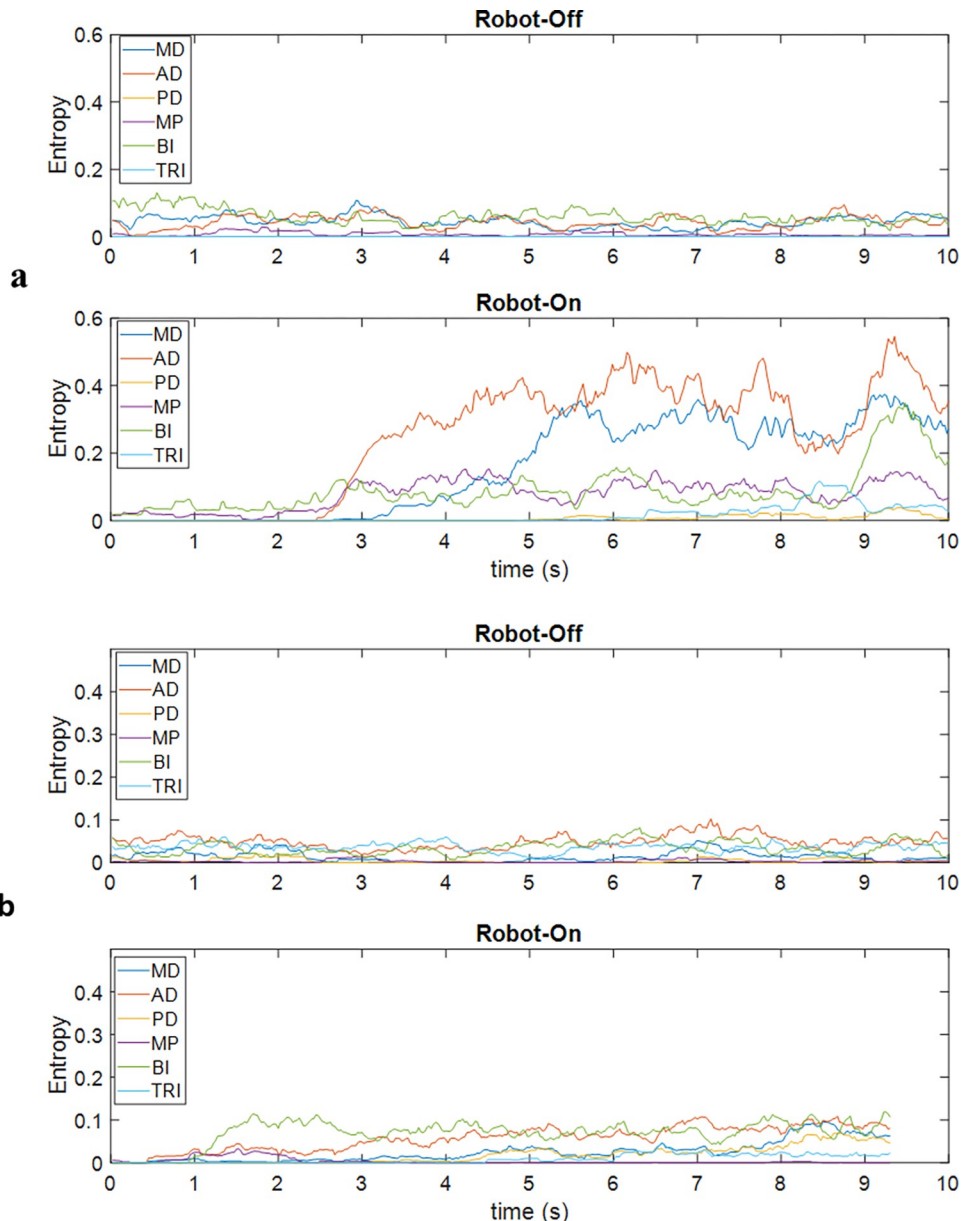

**Fig 5. Entropy from 6 muscles -medium deltoid (MD); anterior deltoid (AD); posterior deltoid (PD); major pectoralis (MP); biceps (BI); triceps (TRI).** (a) From a participant with an FM score of 65 during baseline and training. (b) From a participant with an FM score of 36 during baseline and training.

developed that adapts to user requirements in real-time (and over time) based on synergistic muscle performance, aiding or resisting as needed during progression through robot-assisted training, with the goal of improving outcomes in post-stroke upper extremity rehabilitation. A comprehensive cross-sectional analysis was conducted to evaluate the sensitivity of the aAAN algorithm using data from a single naïve training session in a small sample of individuals with chronic post-stroke hemiplegia.

The algorithm developed in this study used an online adaptive control system based on physiological measures, specifically intending to avoid slacking behavior and improve engagement during training. The robot was used to assist and challenge the user during training,

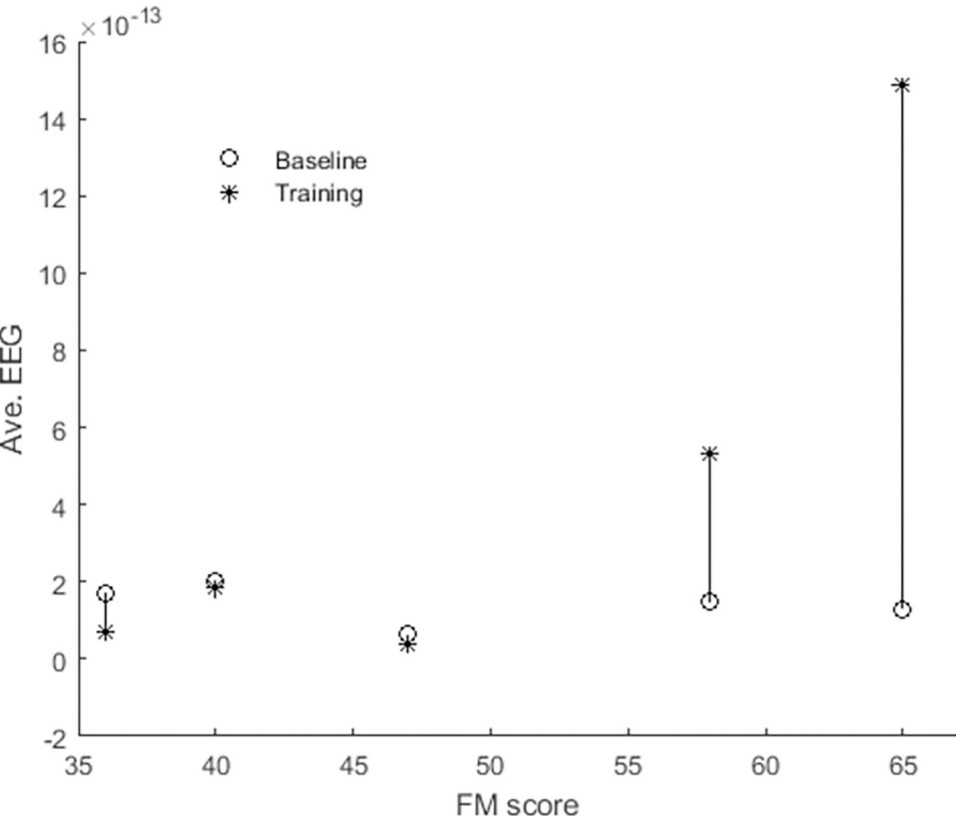

**Fig 6. Mean EEG activity during robot-off phase -baseline- and robot-on–training for all participants.**

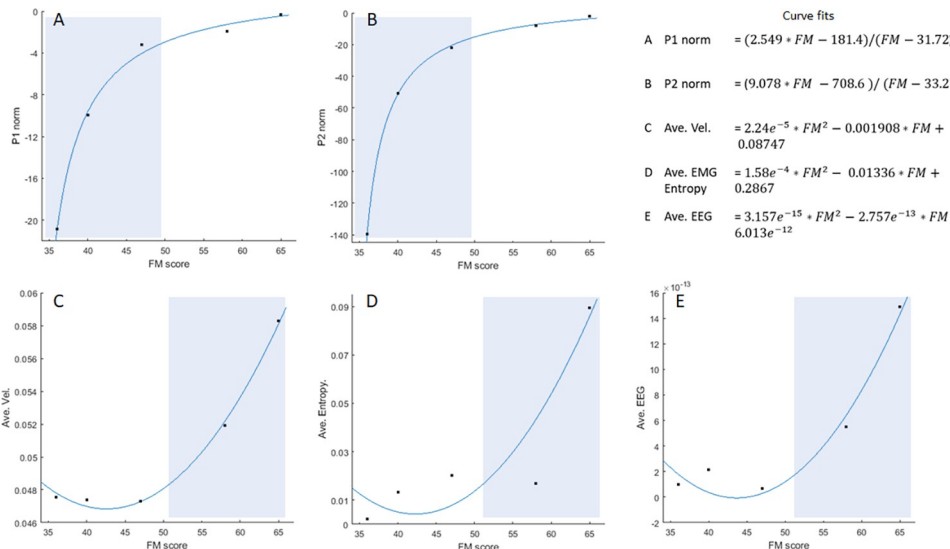

**Fig 7.** Fitted curve for performance parameter FM related to A) $P_{1norm}$; B) $P_{2norm}$; C) Average Velocity; D) Average EMG Entropy; E) Average EEG. The shaded regions show each parameter is sensitive to functional status in the lower FM score region.

keeping them engaged throughout the entire session. To better understand the potential of this new method to impact people with stroke, EMG and EEG was compared between the robot off and on states to test whether training with the aAAN algorithm increases peripheral and central activity, respectively, and the sensitivity of the performance and electrophysiological outcome measures to the user's functional status were evaluated.

Overall, the behavior of the system and how it challenged participants based on their functional status (as shown in Fig 5) was consistent with results presented by Krebs et al. [27] and Benitez et al. [5]. Results showed that the aim of avoiding slacking by providing a challenge at all times was achieved, which can be seen in Fig 5 where the muscle activity increased during training (robot on) when compared to the baseline (robot off) even for the participant with a low score. Such results contradict the results presented by Dipietro et al. [17], where the EMG signal was also used to trigger the robot, and their results showed that when the robot was on, the EMG amplitude reduced for all participants. The authors explain this decrease in EMG as a validation that their robot was assisting the participant in completing the task, but it also indicates that slacking might occur. In contrast, our results showed that slacking was avoided (see Fig 5) when the robot was assisting, which was accomplished by the algorithm adding sufficient challenge when performance was improving (as illustrated in Fig 1).

Another relevant finding was that higher brain activity was observed during training for some participants (Fig 6) with higher FM score, which is expected when more effort is required from the participant [18, 19, 22]. Although there was only a small change for some participants, resulting in a statistically insignificant difference overall, the effect appeared to strengthen with reduced disability. This result suggests that the participant had to do not only more physical work (Fig 5) but more cognitive work (Fig 6) when the robot was on, which may infer that continued training would increases the engagement needed from the participant, potentially improving neuroplasticity in the long term.

It was also evident in Fig 6 that those with lower functional status did not show increases in cortical activity. The results in Fig 7 may be helpful in explaining this observation. Generally, there was a strong relationship between functional status and measurement outcomes, but the relationship was not a simple straight-line linear relationship; rather, it was always a higher-order equation that best fit the data. The curves' shapes were particularly revealing in terms of what might be expected as functional status improves with continued training.

Position parameters $P_{1norm}$ (Fig 7A) and $P_{2norm}$ (Fig 7B) were more sensitive to change in participants with lower FM scores, suggesting that these parameters may be more important to monitor in those with more severe limitations at an earlier stage of motor recovery. Indeed, Fig 6 indicates that participants with lower FM scores were not yet being challenged by the robot system. In other words, the robot was still helping to assist as needed, rather than resist as warranted. The result for velocity was opposite to the position parameters (Fig 7C), being more sensitive to change in participants with higher functional status. This result is consistent with results reported by Mounis et al. [23]. What it may mean in our study is that positional performance may be a prerequisite to achieving a more robust trajectory, which then allows the controller to shift to a more challenging mode of operation.

Results for mean EMG entropy (Fig 7D) and EEG spectral density (Fig 7E), which were also more sensitive to change in participants with higher functional status, appear to corroborate this observation. As reported by Tang et al. [28], EMG entropy is a suitable feature for quantifying changes in neuromuscular function after stroke. In our study, EMG played two roles–it was the central feature that the controller used to decide its future course, and also used as a way to quantify neural drive.

Based on these findings, it may be hypothesized that participants with lower functional status may be expected to improve more in their performance measures before showing

significant changes in muscle and brain activity during a longitudinal study of repeated training sessions. On the other hand, participants with higher functional status and who would have little performance improvement in positional metrics (ie. already performing well, as shown in Fig 4) would show higher muscle and brain activity due to the increased challenge introduced by the aAAN algorithm for these participants.

Future studies will be needed to verify if the novel control system can accelerate motor and functional recovery over time, and perhaps importantly, whether the training algorithm can be used to help those with more severe disability whose functional recovery has plateaued. Our data suggest these individuals need intense focus on position control, which if improved to the point where their movement velocity is also improving, may facilitate increased peripheral and central involvement in movement control of the arm.

There were limitations to the study. Consistent with many previous studies evaluating robot-assisted therapy performance [24, 25, 27, 28] the sample size was small. Nevertheless, participants represented a sufficiently broad range of functioning to enable observation of how people with different levels of disability respond to the training algorithm, revealing the sensitivity of the new aAAN algorithm. The COVID pandemic shutdown prevented a larger sample from being recruited and eliminated the possibility of longitudinal assessments. Study participants were 80% male; a larger study would be needed to have better representation from women with chronic stroke.

## Supporting information

**S1 Checklist. STROBE statement—Checklist of items that should be included in reports of observational studies.**
(DOCX)

## Acknowledgments

The authors thank the staff at the Institute of Biomedical Engineering, Stan Cassidy Centre for Rehabilitation, and Dr Everette Chalmers Hospital, for their assistance in conducting this study. We are indebted to the participants of the study for volunteering their time.

## Author Contributions

**Conceptualization:** Ana P. Arantes, Nadja Bressan, Chris A. McGibbon.

**Data curation:** Ana P. Arantes, Nadja Bressan, Chris A. McGibbon.

**Formal analysis:** Ana P. Arantes.

**Funding acquisition:** Ana P. Arantes, Chris A. McGibbon.

**Investigation:** Ana P. Arantes, Nadja Bressan, Ludymila R. Borges, Chris A. McGibbon.

**Methodology:** Ana P. Arantes, Nadja Bressan, Chris A. McGibbon.

**Project administration:** Ana P. Arantes, Nadja Bressan, Chris A. McGibbon.

**Resources:** Ana P. Arantes, Chris A. McGibbon.

**Software:** Ana P. Arantes.

**Supervision:** Nadja Bressan, Chris A. McGibbon.

**Validation:** Ana P. Arantes, Nadja Bressan, Chris A. McGibbon.

**Visualization:** Ana P. Arantes, Nadja Bressan, Ludymila R. Borges, Chris A. McGibbon.

**Writing – original draft:** Ana P. Arantes.

**Writing – review & editing:** Ana P. Arantes, Nadja Bressan, Ludymila R. Borges, Chris A. McGibbon.

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
