## [Decision Letter · Decision Letter 0]

14 Jul 2023

PONE-D-23-10180Evaluation of a novel real-time adaptive assist-as-needed controller for robot-assisted upper extremity rehabilitation following strokePLOS ONE

Dear Dr. McGibbon,

Thank you for submitting your manuscript to PLOS ONE. After careful consideration, we feel that it has merit but does not fully meet PLOS ONE’s publication criteria as it currently stands. Therefore, we invite you to submit a revised version of the manuscript that addresses the points raised during the review process.

ACADEMIC EDITOR:The paper presents interesting results in the field of robotics rehabilitation under the perspective of control-as needed strategy. I think that the paper could be further improved since Reviewers had rised some major points that have to be carefully taken into accounrt.

We look forward to receiving your revised manuscript.

Kind regards,

Andrea Tigrini, Ph.D.

Academic Editor

PLOS ONE

Journal Requirements:

4.Please include your full ethics statement in the ‘Methods’ section of your manuscript file. In your statement, please include the full name of the IRB or ethics committee who approved or waived your study, as well as whether or not you obtained informed written or verbal consent. If consent was waived for your study, please include this information in your statement as well.

5. We note you have included a table to which you do not refer in the text of your manuscript. Please ensure that you refer to Table 2 in your text; if accepted, production will need this reference to link the reader to the Table.

Additional Editor Comments:

The paper presents interesting results in the field of robotics rehabilitation under the perspective of control-as needed strategy. I think that the paper could be further improved since Reviewers had rised some major points that have to be carefully taken into accounrt.

Reviewers' comments:

Reviewer's Responses to Questions

**Comments to the Author**

1. Is the manuscript technically sound, and do the data support the conclusions?

Reviewer #1: Yes

Reviewer #2: Yes

2. Has the statistical analysis been performed appropriately and rigorously? 

Reviewer #1: No

Reviewer #2: Yes

3. Have the authors made all data underlying the findings in their manuscript fully available?

Reviewer #1: Yes

Reviewer #2: No

4. Is the manuscript presented in an intelligible fashion and written in standard English?

Reviewer #1: Yes

Reviewer #2: No

5. Review Comments to the Author

Reviewer #1: 1. Please explain the type of controller that being utilized for the adaptive AAN.

2. The evaluation of performance metrics is not clears.

3. In table 2, how do you calculate the spasticity score and record the force assist?

Reviewer #2: The research is scientifically sound while having some shortcomings in structuring the paper and some unreasonable conclusions, undiscussed results, and unreferenced statements.

Regarding data availability statement they declared its availability upon request which contradict with the suggested policy by PLOS ONE protocol in such cases.

the manuscript was poorly structured.

I suggest not to reject the paper because the technical method is really interesting. Therefore, I suggest to give the authors a major revision and giving them the opportunity to modify the shortcomings.

6. PLOS authors have the option to publish the peer review history of their article (what does this mean?). If published, this will include your full peer review and any attached files.

Reviewer #1: No

Reviewer #2: **Yes: **Rami Mobarak

---

## [Author Response · Author response to Decision Letter 0]

5 Sep 2023

We thank the Reviewers for their thoughtful and thorough critiques. Below we address each item raised by the Reviewers and where appropriate provide the page and line numbers that correspond to the revisions in the “redline” version of the revised manuscript.

Reviewer #1: 

1. Please explain the type of controller that is being utilized for the adaptive AAN.

In this study, we present a novel approach to training using a robot-assisted system with an adaptive assist-as-needed controller. A control systems method known as “adaptive control” was utilized to design a controller that adapts to the system according to performance parameters – hence termed “adaptive assisted-as-needed” controller. The controller facilitates the completion of tasks for users while providing just enough force to support their efforts and maintain motivation, while also mitigating the potential for complacency. A key feature of our system involves its adaptability, wherein it responds to the user's individual requirements (controller’s parameters) by gradually reducing assistance and increasing task difficulty as the user's performance improves. This tailored approach aims to optimize the training experience and foster engagement through non-monotonic adjustments.

We have revised the Introduction with the above text (Pg 4, lines 76-81).

2. The evaluation of performance metrics is not clear.

We thank the reviewer for pointing this out. In our original manuscript we referred to the hand kinematics and kinetics (movement trajectory P1 and P2 and associated manipulandum forces) as the “performance parameters”. However, in retrospect, we agree that our use of the term “performance” may have been unclear, given that all the measures we are acquiring (muscle EMG, brain EEG, movement trajectories and force) are essentially measures of performance.

We now refer to P1 and P2 (and their normalized quantities) as “position metrics” in the revised manuscript to avoid this confusion. We have also moved the section on Position Metrics to be a sub-section of the Data Analysis section, where, in retrospect, it should have been located in the first place.

 3. In table 2, how do you calculate the spasticity score and record the force assist?

The spasticity score was the sum of Modified Ashworth Scale (MAS) scores across wrist, elbow and shoulder. This was originally included in the footnote of the Table 2, however, we now include this description and a reference for the MAS in the revised manuscript (Pg 12, lines 255-258).

The force assist was recorded directly in the Phantom software device. We added this information to make it more clear in the instrumentation section (Pg 7, line 135-136)

Reviewer #2: 

The research is scientifically sound while having some shortcomings in structuring the paper and some unreasonable conclusions, undiscussed results, and unreferenced statements.

Regarding the data availability statement they declared its availability upon request which contradict with the suggested policy by PLOS ONE protocol in such cases. The manuscript was poorly structured.

I suggest not to reject the paper because the technical method is really interesting. Therefore, I suggest giving the authors a major revision and giving them the opportunity to modify the shortcomings.

We thank the reviewer for their comprehensive critique, and trust that our revision addresses these shortcomings and better presents the methods, results and conclusions. We have also updated our data availability statement to be more specific about data access through the University of New Brunswick data repository. The data will be made publicly available when the study is published.

 Summary: The authors have done interesting work in suggesting an adaptive assist-as-needed method for Upper extremity rehabilitation after stroke. The flow of the suggested method is clear and reasonable. However, there were large shortcomings in the structure of the paper and the discussion and interpretation of some parts of the results as following:

Majors - There is a poor structure in having two sections of ‘’Methods’’ and ‘’ Experimental Procedure’’ with several subsections within each of them making some information redundant in both sections or sparsely distributed although being related to each other. I suggest combining the 2 sections within one section called ‘’Materials and Methods’’ with combining the closely related subsections in the current version into reduced subsections thus connecting related information and avoiding redundancy.

We agree that the flow in the original narrative was somewhat non-traditional. We have followed the reviewer’s suggestion to unify these sections under a more traditional “Materials and Methods” section. In addition, we have reorganized this section to improve the flow and eliminate repetition. 

- Although there was no statistically significant change in EEG spectral density between baseline and training (t=1.18, df=4, p=0.31), the difference trended toward increased EEG activity. ---This supposed finding (trended toward increased EEG activity) is not supported by any figure or table. 

Thank you for the suggestion. We have added a new figure (Fig. 6) that supports our claim that, at least in our small sample of participants, the EEG findings corroborate the other outcome metrics when viewed according to functional status (Pg 15, line 307-309).

 Minors – “Prior studies also do not incorporate adaptation of the controller based on real-time physiological performance that can appropriately scale assistance or resistance to improve engagement and progression.” ---Needs to be referenced with some studies.

 We have added more references to support this statement (Pg 3, line 67). 

–“Lacking, however, was an overall analysis of the subjects’ progress, including muscle and brain activity, which is crucial to monitoring if motor recovery is occurring.” ---Poor sentence context needs rephrasing. 

This phrase was changed to “However, a comprehensive examination of the subjects' advancement of both muscular and cerebral activity, which is a crucial component to assess motor recovery, were absent” (Pg 3, lines 63-654). 

- The hypothesis of the study (“If this approach is effective, we should observe subjects with higher disability at baseline (lower FMA scores) improve more in movement performance between targets, and those with lower disability at baseline (higher FMA) improve more in synergistic muscle and cortical activity.” ) --- must be emphasized at the end of the introduction and not in the control algorithm subsection in the methods.

Although we understand that such a statement might be considered a hypothesis, we state in the Introduction that our goal was explore the sensitivity of the robot training algorithm. The reason this statement appears in the section describing the controller is that the context needed for describing its expected behaviour requires a full description of the adaptive controller. As such we feel this statement is best placed where it was, however, we have reworded the passage (Pg 7, lines 144-147) and we have further clarified our aim in the Introduction (Pg 5, lines 86-89). 

- Participants. This subsection title does not give a clear description of its content. I suggest it to be changed into ‘’ Participants selection criteria ‘’ 

We changed the subsection title as suggested (Pg 8, line 159). 

- The baseline phase was used to collect ROM. The ROM (I suppose Range of motion) needs to be identified in the first use in the manuscript before directly using the abbreviation even if it’s known in the field. 

Indeed, ROM = range of motion. We have made this correction where first mentioned (Pg 10, line 207).

- To calculate the mean adjusted EMG entropy, we first created an index based on the Modified Ashworth Scale summation, The Modified Ashworth scale summation needs to be referenced.

 Thank you, we added the reference for the Modified Ashworth Scale (Pg 12, line 256). 

 - relationship between the participant's FMA scale and performance metrics (average velocity, parameter 𝑃𝑃1 and 𝑃𝑃2), AND baseline. Velocity, time, and position were quantified as performance parameters during the training. AND The performance parameters were also calculated for each move, letter A to letter B being one move, B to C another move, and so on, similar to the process proposed by Krebs et al. [25]. The first performance parameter is the average deviation from the trajectory. 𝑃𝑃1 ….--- Etc : There is confusion in using the performance parameters/metrics as a general term for both velocity, time, position, P1, and P2 or for each part of them separately as being explained in a separate way in the early sections referred to above. 

We agree that the term “performance parameter” was confusing. We now refer to the movement trajectory metrics P1 and P2, and their normalized quantities, as “position metrics” to avoid this confusion (Pg 11, line 224). We have also moved the section on Position Metrics to be a sub-section of the Data Analysis section, where, in retrospect, it should have been located in the first place.

- Data availability statement is unclear and does not refer to how to access the data although it’s important for the journal policy as highlighted in the table. I suggest the authors read the protocol of the data availability statement as listed by the journal. 

We have updated our data availability statement to be more specific about data access through the University of New Brunswick data repository.

- There was little usage of the word ‘’ we ‘’ in the manuscript, in contrast to the major rest of the paper which is in the passive form. This should be unified.

We agree that “we” was used in numerous locations where it wasn’t needed and have thus reworded most sentences to avoid passive language.

- “…Benitez et al. [4]. Although our sample size was small, based on these findings, we are confident the post-stroke sample we recruited represented a sufficiently broad range to evaluate the behavior of the new aAAN algorithm.” --- If there is a previous study that confirms the reliability of the evaluation of such methods for few subjects then it must be cited here, otherwise in science and from a statistical point of view there is no reason for the confidence of the results in a large population when it’s performed on a small sample even if it aligns well with the previous studies cohort size. In fact, the authors already suggested fixing this limitation in future studies.

We agree with the reviewer but point out that what we stated was only that our sample was representative of the stroke population (wide FMA range), which allowed us to observe how stroke survivors with different levels of disability respond to the training algorithm. We have clarified this by moving that statement to the Limitations section where it is given better context (Pg 19, lines 404-407). 

- In Figure 6, The velocity which is recognized by authors as a performance measure showed a different behavior from P1, P2 which are also performance measures while similar behavior to EEG and EMG. Although P1, P2, EEG, and EMG behavior with FMA was well explained by the authors in the discussion part, but they did not provide a reasonable discussion statement about the velocity pattern.

As stated above, we have relabelled these measures to avoid the confusion and ambiguity of the term “performance” as an individual measure. Note that Fig 6 is now Fig 7. We also discuss more fully the velocity finding (Pg 19, lines 397-402).

---

## [Decision Letter · Decision Letter 1]

21 Sep 2023

PONE-D-23-10180R1Evaluation of a novel real-time adaptive assist-as-needed controller for robot-assisted upper extremity rehabilitation following strokePLOS ONE

Dear Dr. McGibbon,

Thank you for submitting your manuscript to PLOS ONE. After careful consideration, we feel that it has merit but does not fully meet PLOS ONE’s publication criteria as it currently stands. Therefore, we invite you to submit a revised version of the manuscript that addresses the points raised during the review process.

We look forward to receiving your revised manuscript.

Kind regards,

Andrea Tigrini, Ph.D.

Academic Editor

PLOS ONE

Journal Requirements:

**Additional Editor Comments:**

Authors have substantially updated the manuscript. However, Reviewers have highlighted minor concerns that have to be solved before publication of the manscript.

Reviewers' comments:

Reviewer's Responses to Questions

**Comments to the Author**

1. If the authors have adequately addressed your comments raised in a previous round of review and you feel that this manuscript is now acceptable for publication, you may indicate that here to bypass the “Comments to the Author” section, enter your conflict of interest statement in the “Confidential to Editor” section, and submit your "Accept" recommendation.

Reviewer #2: All comments have been addressed

2. Is the manuscript technically sound, and do the data support the conclusions?

Reviewer #2: Yes

3. Has the statistical analysis been performed appropriately and rigorously? 

Reviewer #2: Yes

4. Have the authors made all data underlying the findings in their manuscript fully available?

Reviewer #2: Yes

5. Is the manuscript presented in an intelligible fashion and written in standard English?

Reviewer #2: Yes

6. Review Comments to the Author

Reviewer #2: I’d like to thank the authors for their appropriate modifications and clarifications making the context more clear and eliminating the previous confusing points. I would accept the paper to be published only after completing this final minor point :

- Although previously highlighted and the authors said that they managed to solve this point, there's still a lot of usage of ''we'' in the context such as :

'255 drive, we normalized the EMG'

'267 baseline (robot off) and training (robot on) sessions we selected the beta wave range (16 - 24Hz)'

'334 14]. We present a novel EMG-based adaptive AAN algorithm'

'post337

stroke upper extremity rehabilitation. Is this paper we present a comprehensive cross-sectional 338 analysis to...'

'404 robot-assisted therapy performance [24-27] the sample size was small. Nevertheless, we are 405 confident the post-stroke'

7. PLOS authors have the option to publish the peer review history of their article (what does this mean?). If published, this will include your full peer review and any attached files.

Reviewer #2: **Yes: **Rami Mobarak

---

## [Author Response · Author response to Decision Letter 1]

22 Sep 2023

We appreciate the prompt review of our revised manuscript and recognize the Reviewers’ time commitment to assist in further strengthening the manuscript. Below we address each remaining item raised by the Reviewer and provide the page and line numbers that correspond to the revisions in the “redline” version of the revised manuscript.

Reviewer #1: "I’d like to thank the authors for their appropriate modifications and clarifications making the context more clear and eliminating the previous confusing points. I would accept the paper to be published only after completing this final minor point :

- Although previously highlighted and the authors said that they managed to solve this point, there's still a lot of usage of ''we'' in the context such as :"

Response: We have reworded and/or rephrased the sentences below to eliminate “we/us”.

'255 drive, we normalized the EMG' : “…EMG entropy was normalized to an index of spasticity…” (Pg 12 Line 254)

'267 baseline (robot off) and training (robot on) sessions we selected the beta wave range (16 - 24Hz)' : “…the beta wave range (16 - 24Hz) was selected…” (Pg 13 Line 266)

'334 14]. We present a novel EMG-based adaptive AAN algorithm' : “…a novel EMG-based adaptive AAN algorithm was developed that…” (Pg 16 Line 333)

'post337stroke upper extremity rehabilitation. Is this paper we present a comprehensive cross-sectional338 analysis to...' : “A comprehensive cross-sectional analysis was conducted…” (Pg 16 Line 336)

'404 robot-assisted therapy performance [24-27] the sample size was small. 

Nevertheless, we are405 confident the post-stroke' : “Nevertheless, participants represented a sufficiently broad range of functioning to enable observation of how people…” (Pg 19 Line 404-6)

---

## [Editor Report · Decision Letter 2]

25 Sep 2023

Evaluation of a novel real-time adaptive assist-as-needed controller for robot-assisted upper extremity rehabilitation following stroke

PONE-D-23-10180R2

Dear Dr. McGibbon,

We’re pleased to inform you that your manuscript has been judged scientifically suitable for publication and will be formally accepted for publication once it meets all outstanding technical requirements.

Kind regards,

Andrea Tigrini, Ph.D.

Academic Editor

PLOS ONE

Additional Editor Comments (optional):

I'm pleased to inform the Authors that the manuscript can be accepted for publication. All the minor concerns were coherently fulfilled.

---

## [Editor Report · Acceptance letter]

3 Oct 2023

PONE-D-23-10180R2 

Evaluation of a novel real-time adaptive assist-as-needed controller for robot-assisted upper extremity rehabilitation following stroke 

Dear Dr. McGibbon:

I'm pleased to inform you that your manuscript has been deemed suitable for publication in PLOS ONE. Congratulations! Your manuscript is now with our production department. 

Kind regards, 

on behalf of

Dr. Andrea Tigrini 

Academic Editor

PLOS ONE